# At-Sensor Radiometric Correction of a Multispectral Camera (RedEdge) for sUAS Vegetation Mapping

**DOI:** 10.3390/s21248224

**Published:** 2021-12-09

**Authors:** Cuizhen Wang

**Affiliations:** Department of Geography, University of South Carolina, Columba, SC 29208, USA; cwang@mailbox.sc.edu

**Keywords:** sUAS, RedEdge camera, radiometric correction, surface reflectance orthoimage, red edge vegetation index

## Abstract

Rapid advancement of drone technology enables small unmanned aircraft systems (sUAS) for quantitative applications in public and private sectors. The drone-mounted 5-band MicaSense RedEdge cameras, for example, have been popularly adopted in the agroindustry for assessment of crop healthiness. The camera extracts surface reflectance by referring to a pre-calibrated reflectance panel (CRP). This study tests the performance of a Matrace100/RedEdge-M camera in extracting surface reflectance orthoimages. Exploring multiple flights and field experiments, an at-sensor radiometric correction model was developed that integrated the default CRP and a Downwelling Light Sensor (DLS). Results at three vegetated sites reveal that the current CRP-only RedEdge-M correction procedure works fine except the NIR band, and the performance is less stable on cloudy days affected by sun diurnal, weather, and ground variations. The proposed radiometric correction model effectively reduces these local impacts to the extracted surface reflectance. Results also reveal that the Normalized Difference Vegetation Index (NDVI) from the RedEdge orthoimage is prone to overestimation and saturation in vegetated fields. Taking advantage of the camera’s red edge band centered at 717 nm, this study proposes a red edge NDVI (ReNDVI). The non-vegetation can be easily excluded with ReNDVI < 0.1. For vegetation, the ReNDVI provides reasonable values in a wider histogram than NDVI. It could be better applied to assess vegetation healthiness across the site.

## 1. Introduction

In recent years, small unmanned aircraft systems (sUAS) have been utilized in various fields. Mainly for near-surface visual inspection, sUAS are adopted by federal and state agencies for timely surveillance and first response rescue, and standards are released for safe and lawful operational services [1]. sUAS is also defined as “personal remote sensing” [2], and provides quantitative information including tree canopy structure [3], vegetation cover and biomass [4,5] and information concerning the agroindustry [6].

Among the sensors mounted on drones, the most common ones are the low-cost optical cameras that provide centimeter-level imagery in the fields of interest. With this true-color sUAS imagery, a number of RGB indices have been compared for optimal quantitative assessment of vegetation healthiness [6,7,8]. Taking advantage of the red and near infrared light to extract the normalized difference vegetation index (NDVI), NDVI cameras have also been developed in the drone industry for crop scouting and mapping. For example, a drone mapping software, DroneDeploy, subjectively sets the thresholds of NDVI < 0.0 as non-vegetation, 0.0–0.33 as unhealthy plant, 0.33–0.66 as healthy plant, and NDVI > 0.66 as very healthy plant [9,10] and provides an intensive review of a set of drone sensors and vegetation indices used for sUAS vegetation monitoring.

Radiometric accuracy is the premise of the abovementioned biophysical quantification practices. Without effective radiometric calibration, drone imagery heavily suffers from sun illumination affected by time of day, cloud, and other weather conditions. The drone images and drone-extracted vegetation indices are therefore prone to uncertainties in quantitative applications.

More sophisticated multispectral sensors are developed for drone-based measurement. For example, MicaSense RedEdge (MicaSense, Seattle, WA, USA) cameras have been popularly accepted in the drone agricultural industry for functions such as crop health monitoring, weed detection, irrigation, and water management [11]. This multispectral camera contains five bands in the visible-near infrared region with narrow bandwidths (10–40 nm), which are more sensitive to land surface reflection than those low-cost, true-color drone cameras. A recent study reported a tuneable Fabry-Pérot Interferometer (FPI) hyperspectral camera (National Physical Laboratory, Teddington, UK) in the 500–900 nm region with 10–30 nm bandwidth, which showed promising results concerning drone-collected reflectance imagery [12].

These sensors often have their radiometric properties pre-calibrated at the factory in order to correct systematic errors [13]. The laboratory calibration relies on a uniform light source and is subject to certain operating limitations as sometimes reported in the sensor specifications [14]. Sensor functionality may also degrade over time. Therefore, sensors often need to be re-calibrated on a timely basis to ensure proper operation. Some studies conducted their own in laboratory calibration of sUAS sensors. For example, reference [12] tested a procedure of absolute radiometric calibration to directly measure reflectance of a hyperspectral camera. Reference [15] proposed a radiometric calibration method based on relative spectral response of the camera that achieved lower percent errors than its currently adopted procedure.

For real-world deployment, sUAS sensors are vulnerable to vibration and wind effect during flight. Their radiometric properties may also be impacted by other environmental variations such as temperature, solar illumination, and weather conditions. Therefore, even with calibrated sensors, in situ calibration is often needed to improve accuracies of quantitative applications. The most common in situ calibration approach is the Empirical Line Method (ELM) to build the relationships between the reflectance of calibrated targets and their top-of-atmosphere radiance [16]. It requires the real-time reflectance measurements of multiple artificial or natural targets. References [17,18] explored the irradiance-based ELM calibration by considering the aerosol effects in the relationship. This empirical calibration process is often time consuming and not practically efficient.

The real-time field spectra may not be needed when a reference standard is employed in field. The RedEdge system utilizes a pre-calibrated reflectance panel (CRP) set up in field and a Downwelling Light Sensor (DLS) mounted atop a drone to capture the hemispherical irradiance during the flight. The current procedure of RedEdge image correction in common drone data processing packages (e.g., Pix4DMapper) mostly utilizes the CRP panel itself [19]. However, some studies have reported the relatively higher percent errors of its radiometric calibration [15,20]. Our recent practices in a woodland using a RedEdge-M camera [3] also suggested poorly calibrated reflectance, which resulted in overwhelmingly high NDVI of tree canopies.

This study aims to test the radiometric calibration procedure of a RedEdge-M camera utilizing both CRP and DLS sensors during sUAS image acquisition. The in-field spectra and drone calibration images of five panels were collected. A real time, an at-sensor correction model was developed and tested at three study sites. The proposed radiometric correction model could be universally applied for other studies in quantitative vegetation mapping using RedEdge cameras.

## 2. Study Sites, Sensors, and Experiments

### 2.1. Study Sites and Sensor/Panel Properties

This study took advantage of previous sUAS missions in 2020 at three study sites in South Carolina, USA. One is a woodland site at the Sweet Bay Pond Dam, a private earthen dam located around 18 km from downtown Columbia. Trees overgrow into close canopy at the downslope of the dam. The dominant tree species are black gum (*Nyssa sylvatica*), tulip poplar (*Liriodendron tulipifera*), and loblolly pine (*Pinus taeda*). In August–September, multiple sUAS missions were launched to assist dam safety inspection [3]. The second site is a coastal marsh in the North Inlet Estuary, Georgetown. The site is located in an intertidal marsh dominated with high marsh toward the land (up shore) and low marsh in the interior estuary. The dominant species is smooth cordgrass (*Spartina alterniflora*). Four sUAS missions were launched in different areas of the estuary on 30 August 2020, to investigate drone-assisted marsh biomass mapping [5]. The third site is a sports field at River Springs Elementary School in Irmo. It is a piece of lawn field with short, regularly mowed grasses. Three sUAS missions were launched in August–September 2020. Furthermore, the site served as a field experimental site to collect on-ground panel images and spectra in May 2021. Hereafter the three sites are referred to as forest, marsh, and grass, respectively. An oblique-view picture of each site is displayed in Figure 1a as reference. These pictures were taken during each flight.

The sUAS is assembled with a RedEdge-M camera mounted on DJI Matrice100 (M100). The camera has narrow bandwidths: 10 nm for red and red edge bands; 20 nm for blue and green bands (Table 1). The NIR band has a wider bandwidth (40 nm), indicating its lower radiometric sensitivity than other bands. The camera is accompanied with a 10 × 10 cm CRP board, which has a pre-calibrated absolute reflectance curve, by the manufacturer, MicaSense. The CRP is a homogeneous scattering panel at around 50% reflecting ratio in the five RedEdge bands. The CRP photos are taken at 1-m height above the panel prior to and right after a flight mission (as illustrated in Figure 1b). With the bar code embedded on the panel, the CRP reflectance can be automatically recognized in image processing packages. The camera is also assembled with a Downwelling Light Sensor (DLS) on top of the drone facing the open sky (Figure 1b).

Table 1 lists the spectral properties of the RedEdge-M camera and the CRP reflectance provided by the manufacturer. The calibration curve of the CRP is in the range of 400–850 nm at 1 nm interval. For each RedEdge-M band, the CRP reflectance is calculated as the average of the reflectance values at the corresponding bandwidth of the calibration curve (listed in Table 1). Aside from the MicaSense CRP, a ground calibration board made by MAPIR Inc was also tested during field experiments. The MAPIR board contains four diffuse scattering panels in black, dark gray, light gray, and white that have been well calibrated (courtesy of MAPIR CAMERA, San Diego, USA. Communicated on 28 June 2021). Each MAPIR panel is 10.5 × 14 cm in size. Their reflectance values at the designated RedEdge-M bands (also listed in Table 1) are averaged from the manufacturer-provided calibration curves.

### 2.2. Flight and On-Ground Experiments

The sUAS missions in 2020 included five flights at the forest site (August–September 2020), four at the marsh site (30 August 2020), and three at the grass field (August–September 2020). For the marsh site, the low tide in the estuary on August 30 was 12:43 p.m. (0.3 ft) per recordings at the Oyster Landing Station, SC (#8662299). The sUAS images collected in the last flight (at 3:36 p.m.) are affected by the higher tide. Except this flight, all sUAS images and field spectra were collected within two hours from noon to avoid long shadows. For flight missions, the Atlas Flight APP designed for the RedEdge cameras was adopted for autonomous flight and image collection. Flight parameters were set as 85% endlap (along flight path) and 80% sidelap (cross path) at 90 m flight height above ground. The sUAS images were processed in the Pix4DMapper package as well as the open-source MicaSense package available at GitHUB [21].

Two on-ground field experiments (23 and 30 May 2021) were conducted at the grass site, where only calibration images and field spectra of the CRP and MAPIR panels were collected. sUAS flights were not launched on these two dates because only the panel calibration images were needed for validation. The field is covered with regularly mowed short grass (<3 cm high). In each experiment, the two calibration boards (CRP and MAPIR) were laid on grass. Three RedEdge-M photos were taken at 1 m above both targets (as shown in Figure 1). Right after the photo shots, three spectra samples above each panel were collected using a hand-held Jaz spectrometer in a spectral range of 340–1023 nm at 0.3 nm interval. During Jaz spectral collection, the MAPIR white panel served as the reference panel before collecting spectra of other panels. For this white panel, the surface reflectance of each band is listed in Table 1. At the center of each panel, three reflectance spectra were taken in a nadir view and short distance (10 cm above panel) to avoid scattering noises.

Both sunny and cloudy weather conditions were considered in these experiments. Table 2 summarizes the date/time and weather conditions of all experiments. The samples represent the number of the RedEdge-M calibration images utilized in this study. Each calibration image can extract a pair of CRP and DLS irradiance values.

## 3. Methodological Design

### 3.1. RedEdge-M Image Calibration

The RedEdge-M images are recorded in 16 bit. For all RedEdge sensors, a radio-metric calibration model has been developed to convert the raw pixel values into ab-solute spectral radiance. The equation is written as [19]:(1)$$L\left(\lambda \right)=V\left(x,y\right)\times \frac{a_{1}}{g}\times \frac{DN\left(\lambda \right)-DN_{BL}}{t_{e}+a_{2}y-a_{3}t_{e}y}$$
where L is the spectral radiance (W/m^2^/sr/nm) at each RedEdge-M band (*λ*), DN is the raw digital number of a given pixel, and $DN_{BL}$ is the black level value that can be found in metadata tags. Both DN and $DN_{BL}\;$ are normalized to [0, 1] by dividing with 65,536. $a_{1}$, $a_{2}$, and $a_{3}$ are three radiometric calibration coefficients, $t_{e}$ is the image exposure time, and g is the sensor gain setting. For any recorded image, all these coefficients associated with that image are recorded in metadata tags. At any point on the image, x and y are the pixel’s column and row number, respectively. V(x,y) is a radial vignette model to correct the fall-off in light sensitivity at pixel (x,y) when it is out of the image center.
(2)$$V\left(x,y\right)=\frac{1}{1+k_{0}\times r\left(x,y\right)+k_{1}\times r{\left(x,y\right)}^{2}+k_{2}\times r{\left(x,y\right)}^{3}+k_{3}\times r{\left(x,y\right)}^{4}+k_{4}\times r{\left(x,y\right)}^{5}+k_{5}\times r{\left(x,y\right)}^{6}}$$
where *r* is the distance from the pixel to image center $\left(c_{x},\;c_{y}\right)$, with $r\left(x,y\right)=\sqrt{{\left(x-c_{x}\right)}^{2}+{\left(y-c_{y}\right)}^{2}}$. The polynomial coefficients $k_{i}$ (*i* = 0,…,5) are the vignette correction factors that are also recorded in the metadata tags.

### 3.2. RedEdge-M Radiometric Correction

The radiance after calibration is further converted to surface reflectance for spectral analysis. Assisted with a pre-calibrated CRP panel, the surface reflectance (ρ) of the RedEdge-M image can be simply converted as:(3)$$\rho \left(\lambda \right)=\frac{\rho_{CRP}}{L_{CRP}\left(\lambda \right)}\times L\left(\lambda \right)$$
where $\rho_{CRP}$ is the pre-calibrated reflectance of the CRP panel (as listed in Table 1). The $L_{CRP}$ represents the radiance of the CRP panel extracted from Equation (1).

For the RedEdge images collected a drone mission, Equation (3) has been adopted in most sUAS image processing software packages such as Pix4DMapper. These packages scan the QR code and identify a squared area within the CRP panel, then calculate the average radiance from Equation (1). The ratio of the CRP’s reflectance and its average radiance is applied to all pixels of the image to extract surface reflectance.

This approach is easy to use and works well when the relative biophysical quantities are of the major concern, for example, assessing crop stress within a field based on the field-wide NDVI distribution as demonstrated in agricultural case studies [11]. However, the radiometric correction relies on one single panel with a pre-determined constant coefficient for each band. In field, the panel’s reflectance may be affected by solar illumination and path scattering that vary with time and atmospheric and environmental conditions. Therefore, a more rigid radiometric correction method is needed to better calculate the reflective values of the acquired RedEdge-M images.

### 3.3. At-Sensor DLS Radiometric Correction

As described above, the RedEdge-M camera takes the CRP calibration images on ground before and after a flight mission. With Equation (1), the image records the radiance reflected out of the CRP panel, which can be transformed to the incoming irradiance of each band given the known reflectance of the diffuse material. At the same time, the sUAS is assembled with a Downwelling Light Sensor (DLS) atop of the drone facing the open sky (see Figure 1). The DLS records the incoming irradiance in each band at a unit of W/m^2^/nm. The irradiance recordings are stored as metadata of each sUAS image. Influenced by the mechanical differences between the two sensors and environmental scattering, the two irradiance recordings would not be identical but should follow a linear relationship:(4)$$Irr_{CRP}\left(\lambda \right)=a\times Irr_{DLS}\left(\lambda \right)+b$$
where $Irr_{CRP}\left(\lambda \right)$ and $Irr_{DLS}\left(\lambda \right)$ represent the camera-recorded irradiance reaching the CRP and the DLS-recorded irradiance atop of the drone. Note the DLS is installed upward facing the direct sky. At flight altitude, path scattering from surrounding environment is limited. On the other hand, the CRP is laid on ground that is inevitably impacted by path scattering. The coefficients *a* and *b* fairly explain the differences between the two sensors.

The CRP is made of diffuser materials that are supposed to have equal radiance in all directions. The $Irr_{CRP}\left(\lambda \right)$ can thus be simplified as the CRP radiance multiplying pi (π) [21]. Similarly, the DLS radiance is calculated as the recorded irradiance dividing with π. Moreover, the CRP is built with known reflectance coefficients $\rho_{CRP}\left(\lambda \right)$, which determines its reflected radiance. Equation (4) is converted to:(5)$$\frac{L_{CRP}\left(\lambda \right)}{\rho_{CRP}}=a\times L_{DLS}\left(\lambda \right)+\frac{b}{\pi }$$

Note $L_{CRP}\left(\lambda \right)$ in Equation (5) is the reflected, at-sensor radiance of the CRP panel. Since the DLS radiance is less affected by upwelling radiation, it can replace the CRP radiance in Equation (3):(6)$$\rho_{cor}\left(\lambda \right)=\frac{\rho_{CRP}}{L_{DLS}\left(\lambda \right)\times \rho_{CRP}}\times L\left(\lambda \right)=\frac{L\left(\lambda \right)}{L_{DLS}\left(\lambda \right)}$$
where $\rho_{cor}\left(\lambda \right)$ represents the corrected reflectance of the RedEdge-M image. In the denominator, $\rho_{CRP}$ is used to convert the incoming radiance to reflected radiance out of the panel. In Equation (6), replacing $L_{DLS}\left(\lambda \right)$ with that in Equation (5) and L(λ) with that in Equation (3), the CRP-based image reflectance can be corrected as:(7)$$\rho_{cor}\left(\lambda \right)=\frac{a}{1-\frac{b*\rho_{CRP}}{\pi *L_{CRP}\left(\lambda \right)}}\times \rho \left(\lambda \right)$$

Equation (7) reveals that image reflectance can be linearly corrected from the CRP-based reflectance ρ(λ) extracted in the original procedure. The correction factor for each flight mission is thus defined as:(8)$$Cor\left(\lambda \right)=\frac{a}{1-\frac{b*\rho_{CRP}}{\pi *L_{CRP}\left(\lambda \right)}}$$

Among the coefficients in calculating the correction factor in Equation (8), the CRP reflectance $\rho_{CRP}$ of each band is pre-determined and provided by the manufacturer. The averaged CRP radiance $L_{CRP}$ is mission specific and is extracted from the panel calibration image of a flight mission. Therefore, the only question to be solved is to get the coefficients *a* and *b* from the CRP-DLS irradiance regression.

Both CRP and DLS are made of diffuse materials and therefore possess a common relationship between their irradiance recordings. This study relies on all calibration data collected in multiple flights at different study sites to build the CRP-DLS relationship. Details are described in the following section.

### 3.4. Data Analysis

For each calibration image, with Equations (1)–(3) the CRP radiance was retrieved, and the reflectance image was extracted using GitHub open-source package, micasense/Imageprocessing [21]. The DLS radiance was retrieved from the metadata stored in the image. The correction factor (Equation (8)) was calculated that can be applied to all RedEdge images in the same mission. Figure 2 demonstrates the areas of interest (AOIs) of the CRP panel (Figure 2a) and the four MAPIR panels (Figure 2b). All pixels within the AOI are averaged to extract the average panel radiance. The CRP panel is automatically recognized in the GitHUB open-source package based on its barcode, while the AOIs of the MAPIR panels are manually selected in each image. Surprisingly there is an “O” shape in the MAPIR white panel that is invisible to the naked eye.

The corrected RedEdge-M reflectance was validated with field spectra from the Jaz spectrometer. The field spectra served as ground truth to validate the RedEdge-M reflectance. Be aware that the “O” shape on the MAPIR white panel may introduce noises in the validation process.

The sUAS images in each flight mission were processed in the Pix4DMapper package to extract the surface reflectance orthoimage. The original radiometric correction method (Equations (1)–(3)) is adopted in the package. With the proposed at-sensor correction model (Equation (7)), the surface reflectance orthoimage is further corrected. The results at different sites were compared in this study.

For quantitative analysis, two vegetation indices were extracted. The normalized difference vegetation index (NDVI) has been most popularly applied in the remote sensing community:(9)$$NDVI=\frac{\rho_{NIR}-\rho_{Red}}{\rho_{NIR}+\rho_{Red}}$$
where $\rho_{NIR}$ and $\rho_{Red}$ are surface reflectance of NIR and red band, respectively.

Taking advantage of the red edge band of the RedEdge-M camera, we also propose a red-edge NDVI, hereafter defined as ReNDVI:(10)$$ReNDVI=\frac{\rho_{RedEdge}-\rho_{Red}}{\rho_{RedEdge}+\rho_{Red}}$$

Similarly, $\rho_{RedEdge}$ is the surface reflectance of the red edge band. Note the ReNDVI simply replaces the NIR band with the red edge band in the formula.

## 4. Experimental Results

### 4.1. The CRP and DLS Radiometric Characteristics

When acquiring the RedEdge-M images of the CRP panel, solar radiation can be recorded with both CRP and DLS. The CRP radiance is extracted with Equation (1). The DLS records solar irradiance directly and stores the information in the metadata.

South Carolina has a subtropical climate with hot, humid summers. Even on sunny days, thin clouds start to accumulate in late morning due to a high concentration of water vapor in the atmosphere, which is especially the case at coastal marsh sites. We categorize all sUAS images according to the weather conditions when they were acquired: Sunny and cloudy. However, only two experiments had pure sunny conditions. Other sunny days were actually sunny with thin cloud, which inevitably affected the recorded solar irradiation and atmospheric path scattering.

For each RedEdge-M band of the calibration images, the irradiance values of all samples were plotted in accordance with cloudy or sunny days (Figure 3a). Reasonably, both CRP and DLS recorded much higher irradiance on sunny days and apparently lower irradiance on cloudy days. For example, in the field experiment on 30 May 2021, it started in an overcast condition at noon, then clouds gradually reduced. At the last sample at 13:15 p.m., it became sunny with cloud patches. Therefore, the irradiance recorded in both CRP and DLS dramatically increased from about 0.5 W/m^2^/nm at the first sample to around 1.8 W/m^2^/nm at the last sample.

The CRP panel recorded higher irradiance than DLS on both sunny and cloudy days (Figure 3a). The marks of one standard deviation are added to the curves in the figure. The deviation between the two sensors is larger on sunny days. Irradiance on sunny days is dominated with direct sunlight. However, thin cloud patches were present on most sunny days of our experiments, which may be attributed to the higher deviation observed in the figure. Irradiance on cloudy days is dominated by skylight, or path radiation in the atmosphere. Aside from cloud effects, solar radiation also varies with diurnal cycle and geographic locations, which introduces further variations to the recorded irradiance values in the figure.

Both curves show a reasonable decreasing trend along the visible-NIR electromagnetic spectrum. On cloudy days, the irradiance from the two sensors becomes similar and holds a subtle decreasing trend along the spectrum. It should also be noted that, among the five RedEdge-M bands, the green and NIR band reveal the largest deviation between the two sensors. This indicates a stronger correction in these two bands.

We further compared the DLS irradiance recorded at ground (calibration images) and that recorded during a flight (at-flight images). For each flight mission, an at-flight RedEdge-M image with acquisition time close to the ground calibration image was randomly selected for the comparison. In Figure 3b, the DLS irradiance values of all five bands are close to the 1:1 line. Similarly, points at the lower end (cloudy days) have better agreement and those at the higher end (sunny days) are more dispersed. The five points circled in Figure 3b are recordings of one sample collected in a forest flight on 26 August 2020. Clouds began accumulating right after the flight was launched. The exceptional deviation of these five points from the 1:1 line reflects the effect of rapidly changing cloud.

Overall, the sUAS flight missions are short, within 10–15 min, and the ground measured DLS (right before or after the mission) matches well with at-flight DLS recordings. Therefore, the DLS recordings at the calibration image could serve as valid irradiance source for radiometric correction.

### 4.2. At-Sensor RedEdge-M Radiometric Correction

The correction factor, Cor(λ), is related to the ratio between the DLS and CRP irradiance on calibration images. Irradiance samples on sunny and cloudy days have different value ranges but follow a linear relationship in each band. Therefore, a universal DLS-CRP regression over all weather conditions can be achieved. A total of 107 samples were included to build the regression of each RedEdge-M band. All linear regressions in Figure 4 are strongly significant with R^2^ in a range of 0.995–0.997. The regression slopes are greater than 1.0 because the DLS recorded irradiance is lower than the CRP-converted irradiance in all weather conditions (as shown in Figure 3a above).

The coefficients used to extract the correction factor for each band are listed in Table 3. With the universal coefficients, a, b, τ, and $\rho_{CRP}$ determined in this study, the at-sensor correction can be performed concerning all sUAS images in each flight mission by extracting the mission-specific CRP radiance from the calibration panel image. The correction factor is thus determined for each mission.

The correction power varies with the RedEdge-M bands and weather conditions (Figure 5). For all samples in both field and drone experiments, the at-sensor corrected RedEdge-M reflectance is always higher than the original reflectance with a correction factor in a range of 1.01–1.35. The NIR band receives the highest correction, followed by the green band. This indicates the impact of scattering from surrounding vegetation. The blue band needs the least correction. Weather conditions may also affect surface reflectance from RedEdge-M imagery. Figure 5 indicates that RedEdge-M reflectance on cloudy days receives higher correction (larger factor) than on sunny days. It should also be noticed that, especially for Red and NIR bands, the correction factor on cloudy days has much higher deviation (large IQR) from the median, which may be related to less stable radiation properties from the moving clouds. Without correction, sunny days are favored to collect high quality RedEdge-M reflectance images. Our proposed at-sensor correction fairly compensated for the atmospheric effects and therefore could be used in both weather conditions.

### 4.3. Validation with Jaz Field Spectra

At the two on-ground field experiments, the Jaz measured and RedEdge-M reflectance values of all calibration panels (CRP and MAPIR) were compared (Figure 6). For each panel, 10 reflectance points (six in sunny and four in cloudy) were collected by each sensor. Each point represents the measurement at each time interval as listed in Table 2. At each point, the Jaz reflectance was the average of three Jaz spectra at the corresponding bandwidth. Similarly, three panel images were randomly selected, and the RedEdge-M reflectance of each band was averaged across the AOI on these panel images. Both the raw (before correction) and corrected reflectance of RedEdge-M are compared in the figure.

Since the MAPIR white panel was used as the reference, it had consistent Jaz reflectance in each band while the RedEdge-M reflectance varied. For the CRP panel, the RedEdge-M reflectance had constant values (pre-calibrated) in each band while the Jaz reflectance varied. For other MAPIR panels, the black has an average pre-calibrated reflectance of 0.02, dark grey has 0.21, and light grey has 0.27 as marked in Table 1. At these three panels, both Jaz and RedEdge-M reflectance of the calibration panels varied when measured on different dates, affected by local variations such as solar illumination and slight change of the Jaz sensor’s view angle off the desired nadir direction. From low to high along the axis in Figure 6, the panels are MAPIR black, dark grey, and light grey. The MAPIR light gray and dark gray panels have similar reflectance in nature, and their corrected RedEdge-M reflectance values become clustered in the scatterplots.

As shown in Figure 6, the point-to-point values of Jaz and RedEdge-M measurements at the three MAPIR panels distribute along the 1:1 line for both the raw and corrected reflectance. Agreeing with Figure 5 above, the RedEdge-M reflectance in blue band has the best fit with the Jaz reflectance. The corrected and raw measurements are similar, indicating the good quality of the camera in this band. For other bands, the corrected RedEdge-M reflectance is closer to the 1:1 line than the raw data, especially in the higher-reflectance samples. In the NIR band, the corrected reflectance is visually higher than the Jaz reflectance for the lower-reflectance samples. A similar phenomenon occurred for the raw reflectance. This could be attributed to path scattering in the vegetated field. Recall that the Jaz measurements were collected at 10 cm above the panels, while the RedEdge-M CRP images were taken at 1 m above ground. Scattering from surrounding grasses may result in higher reflectance at the MAPIR black and dark gray panels. This low-reflectance noise could be reasonably removed by adding the black panel as the reference in the correction procedure. However, this is beyond the interest of this study, which focuses on real-time, at-sensor correction using the CRP panel itself. The common measures of errors, for example, the coefficient of determination (R2) and the root-mean-square errors (RMSE), are similar between the raw and corrected reflectance because the correction is simply a coefficient multiplied to the raw value. The correction factor varies with calibration image.

The RedEdge-M CRP and MAPIR white panels cannot be directly evaluated in Figure 6 because one has constant horizontal (CRP) and the other has constant vertical (MAPIR white) values in the scatterplots. Table 4 lists the average measurements of each panel in comparison with the pre-calibrated reflectance. The “all-weather” represents the measurements using all 10 samples on both sunny (six samples) and overcast (four samples) days. For the MAPIR white panel, the camera reflectance was lower than the pre-calibrated Jaz reflectance (with an average of 0.82). The corrected reflectance was higher than the raw values and closer to the pre-calibrated values, indicating the improvement of camera performance after correction. The standard deviations of the all-weather measurements reflect the weather effects. Agreeing with previous observations, the camera performance was better in sunny days than overcast. For the RedEdge-M CRP panel, the Jaz measurements generally agree with its pre-calibrated reflectance (with an average of 0.49). Its relatively higher standard deviations indicate that Jaz is prone to weather effects, i.e., lower reflectance on sunny days and higher in overcast conditions. Given the imperfectness of Jaz measurements in these experiments, a more rigid field champaign may be needed for advanced assessment of RedEdge-M performance.

### 4.4. At-Sensor Corrected RedEdge-M Surface Reflectance

One correction factor was calculated for each mission. Three missions at different vegetation sites were explored here to demonstrate the correction process. All sUAS missions were launched in the growing season in 2021: Marsh on 30 August, grass on 27 September, and forest on 22 September. Table 5 lists the correction factors of each RedEdge-M band in the three missions. Agreeing with the regression slopes, the blue band receives the least correction, followed by red edge and red. The green band receives higher correction than other visible bands. The NIR always receives the highest correction among all RedEdge-M bands. The correction also varied with weather conditions. Missions in cloudy weather receive higher correction than in sunny conditions.

Three example single-shot sUAS images were randomly selected to represent the three vegetation types. Each RedEdge-M image has a dimension of 1280 × 960 pixels. For visual comparison, their example surface reflectance images in red and NIR bands are displayed in Figure 7. Healthy vegetation has high concentration of chlorophyll content. Its spectral properties are characterized with strong absorption in red and strong reflection in NIR bands. As shown in Figure 7, the red reflectance is dramatically lower than NIR in all three vegetation types. In comparison with full-cover grass and forest, salt marsh has sparse cover and is less green in field. Therefore, its NIR reflectance is much lower than those of grass and tree. The forest image was taken on sunny days. Tree shadows in between canopies are clearly visible, characterized by low reflectance in both bands.

Table 6 summarizes the minimum and maximum reflectance of each image in Figure 7. The grass image contains both lawn field and concrete driveway. The car on the driveway is also highly reflective. Therefore, the image has the largest min-max range in all RedEdge-M bands. The marsh image has a few pieces of woody boardwalk, which provides a narrower but still reasonable min-max range. The forest image is collected in a woodland almost fully covered with tree canopies. Its min-max range in visible bands is much smaller than the other two images. As highlighted (in bold) in Table 6, the minimum reflectance in the visible bands of the forest image is extremely due to shadow effects. On the other hand, thanks to the majority pixels of healthy trees, the maximum NIR reflectance is exceptionally high (0.6762). Even the red edge band possesses a much higher maximum reflectance (0.2635) than the visible bands.

The reflectance of all land objects has a theoretical range of [0, 1]. As shown in Table 6, the min-max reflectance values of all three vegetation images are in the low end of this full range. This indicates the necessity of contrast enhancement for optimized visual display of RedEdge-M reflectance images. To maintain the white color balance, we chose the method of histogram normalization for contrast enhancement. For each display, the min-max thresholds of 2.5% and 97.5% along the histogram are calculated using all pixels of the three bands. By removing the outliers in both tails of the histogram, the normalization effectively reduces the impact of extremely darker and brighter pixels to the resulted color contrast. In Figure 8, all images are visually pleasant after histogram normalization. The RGB displays reveal the reasonable green tone for vegetation. Vegetation with higher biomass (e.g., forest) has a greener tone in RGB and a more reddish tone in CIR than that with less biomass (e.g., marsh).

### 4.5. Corrected Orthoimages and Vegetation Indices

The at-sensor radiometric correction was performed concerning the surface reflectance orthoimages exported from Pix4DMapper. Figure 9 reveals the RGB and CIR composites of marsh (a), grass (b), and forest (c) of the three missions. Similarly, the normalization enhancement was performed for each display. Both displays reflect the greenness of vegetation across each site. In Figure 9a, high marsh dominates the estuary close to the up shore in the northwest. High marsh is short, sparse, and has much lower greenness than low marsh in transition to the tidal channels to the southwest. In Figure 9b, we could visually identify the uneven greenness of grass on both sides of the driveway. In Figure 10c, the forest mission was launched in early fall (22 September 2020), when black gum trees show early signs of fall color and leaf-off while tulip poplar remains green [3]. We could easily differentiate these tree species from drone orthoimages.

The CIR is known to better reflect vegetation healthiness conditions in a reddish tone. Figure 10a, for example, has a much brighter reddish tone in the tidal low marsh than the sparse high marsh zone where the boardwalk is located. Interestingly, along the wooden boardwalk, meter-sized plots of fertilizer-treated spartina stand out of the low marshes nearby, which has been further studied in our previous research [5]. As shown in the RGB composites, grass (Figure 9b) and forest (Figure 9c) have higher greenness than marsh. In the CIR composites, they become visually saturated in an exceptionally red tone.

Both NDVI and ReNDVI were extracted to further assess the capability of the drone/RedEdge-M system on quantifying vegetation healthiness conditions (Figure 10). At all three sites, the NDVI values are much higher than what we expect in a natural environment. In Figure 10a, the dominant high marsh has abnormally high NDVI values in a range of 0.4–0.65. The treated spartina plots along the boardwalk are nicely distinguished with their NDVI values higher than 0.65. The tidal low marsh in the southeast end even becomes saturated (NDVI > 0.85), which is rare in coastal wetlands. On the other hand, the ReNDVI takes advantage of the red edge band that results in a more reasonable range. The ReNDVI at the marsh site reveals reasonable distribution of vegetation index, with the high marsh holding NDVI in 0.2–0.3 and low marsh in 0.3–0.5 and the dense clusters having ReNDVI > 0.5. The ReNDVI map also reflects a reasonable range of non-vegetation surfaces: Less than 0.1 on the sand-paved road, and a low range of 0.1–0.2 on the mudflat and open water in tidal channels.

The overestimation problem is more obvious at the grass site (Figure 10b) and especially the forest site (Figure 10c). It is not realistic that short grass in a mowed lawn field (as illustrated in Figure 1) has its NDVI in a range of 0.7–0.97. The roadside trees in Figure 10b are mostly longleaf pine and the woodlands in Figure 10c are dominated with deciduous trees [3]. In the acquisition time of early fall, it is less possible that these trees have saturated NDVI (close to 1.0) at both sites. In contrast, the ReNDVI maps reveal reasonable ranges of grass (0.2–0.5) and forest (0.5–0.85). Similarly, the ReNDVI of non-vegetation is in agreement with our general understanding: ReNDVI < 0.1 (mostly negative) for concrete, paved road, and open water at both sites.

The histograms of both indices are displayed along the legend in Figure 10. At each site, the two indices share similar histograms, yet the ReNDVI has lower values than NDVI. Excluding the non-vegetation pixels, all histograms are normally distributed. At the grass and forest sites, the ReNDVI histograms are wider than the corresponding NDVI ones. This indicates that ReNDVI visually reduces the saturation problem as shown in NDVI.

## 5. Discussion

Quantitative applications of drone deployment rely on standardized procedures and products. The RedEdge-M camera takes advantage of a pre-calibrated panel (CRP) for real-time radiometric correction by assuming its constant reflectance in all environments. Exploring a number of flight missions in vegetated fields, this study found that the CRP-based correction in the currently adopted RedEdge-M processing procedure could fairly extract the surface reflectance orthoimages. However, the performance was less stable on cloudy days than sunny days. Multiple field experiments revealed that the measured surface reflectance of the CRP panel varied practically due to varying solar, weather, and other local conditions. A similar phenomenon was observed for the MAPIR panels coming with constant pre-calibrated reflectance in its black, dark/light grey, and white panels.

This study proposes an at-sensor correction approach to leveraging these local effects on surface reflectance in each flight mission. The DLS of the RedEdge-M system directly records the incoming irradiance and therefore it explains the real-time atmospheric conditions. The correction factor is calculated by adjusting radiometric calibration using the DLS irradiance in each mission. At each RedEdge-M band, the correction factor on sunny days is relatively stable, while on cloudy days the correction factor varies largely among different missions, reflecting the unstable irradiance from moving clouds. Integrating the CRP and DLS, the at-sensor correction effectively reduces the atmospheric impact to the extracted surface reflectance orthoimages.

The RedEdge cameras are designed for real-time calibration from the CRP panel in field. Current literature on testing its radiometric correction has been limited. We found one study proposed a similar yet simpler approach named At-Altitude Radiance Ratio (AARR), which basically defined the ratio of the DLS and CRP radiance as the correction factor [20]. As shown in Figure 4 of our study, the relationship between DLS and CRP radiance in each band is not a simple ratio; it contains a small yet positive intercept value. The intercepts are especially not ignorable in red and NIR bands. Sample points in the figure also demonstrate that, in certain circumstances, the DLS and CRP readings could deviate further away from the linear regression line. For these images, the correction based on the DLS/CRP irradiance ratio [20] could introduce high errors into the extracted reflectance. Our approach establishes a universal DLS-CRP relationship by embracing multiple weather and environmental conditions and therefore fairly reduces these local impacts on radiometric correction.

The at-sensor correction is band specific. Considering both sunny and cloudy days, the blue band receives the least correction (close to 1.01) while the NIR band has the highest (>1.25). Given that all study sites are vegetated, reflecting higher NIR than visible light, it is reasonable that the CRP and DLS-extracted NIR reflectance has higher dispersion. The relatively wider NIR bandwidth (40 nm) than the visible and red edge bands (10–20 nm) may also introduce higher uncertainty. Another possible impact may come from the sensitivity of the DLS diffuser. The first-generation DLS does not consider the Fresnel Effect [19], i.e., the diffuser reflects more light at the large sun-sensor angle. For our data processing, a Fresnel Coefficient is calculated to compensate for the sun-sensor angle effect from sun diurnal and seasonal changes, or simply drone pitch and roll [21]. However, uncertainties could remain. For newer RedEdge-MX cameras, the second-generation DLS-2 takes this effect into consideration, which may improve the DLS sensitivity and lead to the improved at-sensor correction of surface reflectance products.

The commonly applied ELM calibration methods require the field-collected reflectance of ground targets for both calibration model and validation. Our proposed calibration method corrects the CRP-calibrated reflectance from the RedEdge cameras and therefore the in-field reflectance is not needed for calibration. One limitation of this study, however, is our incapability of validating the in-flight sUAS imagery at the time this research was investigated. For validation, five calibration panels (CRP and four MAPIR) were employed and their on-ground spectra and RedEdge images were collected. The panels are small in size. Impacted by white paper on both CRP and MAPIR boards (as shown in Figure 1), our experiments found it difficult to identify the pure pixels of these panels on the in-flight drone images at 80–120 m flight height. Relying on the low-cost MAPIR white panel as reference, the recorded spectra from the Jaz spectrometer may also be questionable. For advanced validation of the proposed approach, a more rigid field experiment will be conducted to match in-flight RedEdge reflectance with field spectra on multiple, relatively homogeneous land covers. A high-quality diffuse reflectance panel will be sought (e.g., the Ocean Optics Reflectance Standard with >95% reflectivity) to replace the MAPIR white panel for reference readings. Masking the white paper on the calibration boards with black materials may also reduce its impact to panel reflectance on the sUAS imagery.

With 10–20 nm narrow bands (40 nm for NIR), the RedEdge-M images reveal high visual quality. For quantitative analysis, however, the commonly used NDVI from the red and NIR bands is apparently overestimated. This finding agrees with past studies utilizing the same cameras. For example, in an sUAS experiment for RedEdge calibration [15,20], the RedEdge imagery was collected at a flight height of 375 ft on November 2, 2017, in Henrietta, NY, USA. The NDVI reflectance-extracted in a roadside grass field, similar to our grass site in this study, was in a range of 0.8–0.9, and was even higher on cloudy than sunny days [20]. Be aware that the imagery was acquired at the end of the growing season in a mid-latitude area (43.04° N, 77.70° W). The NDVI values apparently overestimate the greenness of natural grass. The overestimation performance was similar for RedEdge-3 and RedEdge-M [15] cameras. It is not a big concern when the end users only need to visually compare the NDVI differences of a field in one single flight, as demonstrated in the MicaSense brochure [11]. However, it is not practical to directly utilize the RedEdge NDVI for quantitative vegetation mapping.

Practically, this study develops a red edge involved NDVI, or ReNDVI, by replacing the NIR with the red edge band for better practices of vegetation assessment. Note the formula of our proposed ReNDVI is different from the Normalized Difference Red Edge index (NDRE), which has been recognized in the sUAS community [15,22]. The NDRE combines NIR and red edge bands in the calculation. Our tests indicate that the NDRE values at all three sites are much lower and cannot effectively represent vegetation abundance. As shown in the typical reflectance curve of vegetation [23], the RedEdge camera’s red edge band (712–722 nm, centered at 717 nm) is located in the upper range of the red-NIR slope. Its spectral difference with the red absorption band (663–673 nm, centered at 668 nm) is actually larger than with the NIR band (820–860 nm, centered at 840 nm) where the NIR reflectance becomes saturated. Therefore, the proposed ReNDVI has higher values than NDRE, and represents reasonable biophysical properties such as abundance and healthiness in vegetated lands.

The red edge band of the RedEdge cameras has much narrower bandwidth (10 nm) than the NIR band (40 nm) centered at 840 nm. In our experiments, it has much less deviation between the CRP calculated and DLS recorded irradiance values and bears less correction on both sunny and cloudy days. Our results agree with past studies [23], which indicate that a red edge spectral band between the red and NIR bands of the Landsat8 OLI has a high potential to improve agricultural green leaf index (LAI) retrieval [24], and also confirm that the experimental LAI has the best linear correlation with the band combination of the 675 nm and 712 nm that is in alignment with our proposed ReNDVI.

The red edge band is becoming more popular in satellite remote sensing. The multispectral instrument (MSI) on board the Sentinel-2 twin satellites has three red edge bands available—705 nm, 740 nm, and 783 nm—which is expected to boost the red-edge data analysis and utilization. The radiometrically corrected RedEdge cameras mounted on drones could provide valuable ground truth of surface reflectance for Sentinel-2 applications. Taking advantage of these new data sets, further investigation of the red-edge indices will be conducted to advance its applications in fields such as the agroindustry and forestry management.

## 6. Conclusions

This study tests the performance of a RedEdge-M camera in extracting surface reflectance orthoimages and vegetation index products for drone-assisted vegetation mapping. For each sUAS mission, a real-time, at-sensor radiometric correction procedure is established by interactively utilizing the CRP calibration panel and the built-in DLS sensor. The correction effectively leverages the variation of incoming irradiance from sun diurnal, seasonal, and other local effects. The correction power varies in RedEdge-M bands, with the NIR bearing the highest correction in our experiments at vegetated sites. Finally, with the extracted surface reflectance, the proposed ReNDVI reduces the overestimation problem in the original RedEdge-M NDVI. It could easily mask out the non-vegetation pixels (ReNDVI < 0.1) and could be better applied for quantitative analysis of vegetation healthiness. Overall, this study proposes an all-in-one operational procedure to deploy drone-mounted RedEdge cameras in best practices: Flight mission, image radiometric calibration, at-sensor atmospheric correction, normalization enhancement for visual interpretation, and ReNDVI for quantitative health assessment.

## Figures and Tables

**Figure 1 sensors-21-08224-f001:**
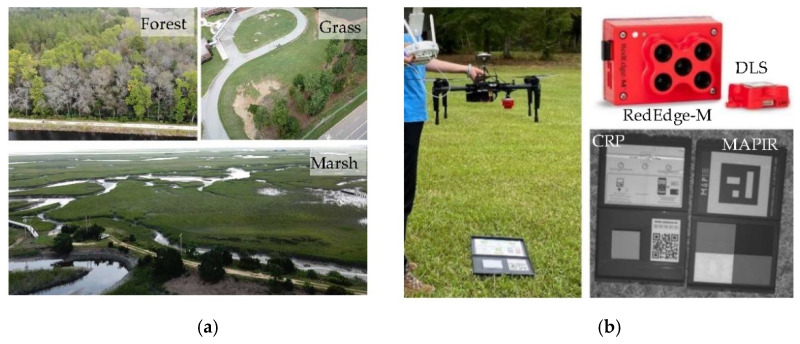
The oblique view of the three sites (forest, grass, marsh) (**a**) and the RedEdge-M camera mounted at the bottom of M100 and the two panels (**b**). The DLS is on the top of drone. Two calibration boards, MicaSense CRP and MAPIR panels (black/dark grey/light grey/white), are used in the study.

**Figure 2 sensors-21-08224-f002:**
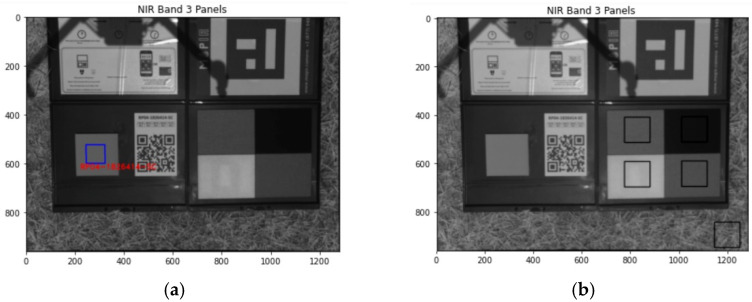
The extracted AOIs of the CRP (**a**) and MAPIR panels (**b**) on one example calibration image (NIR band only).

**Figure 3 sensors-21-08224-f003:**
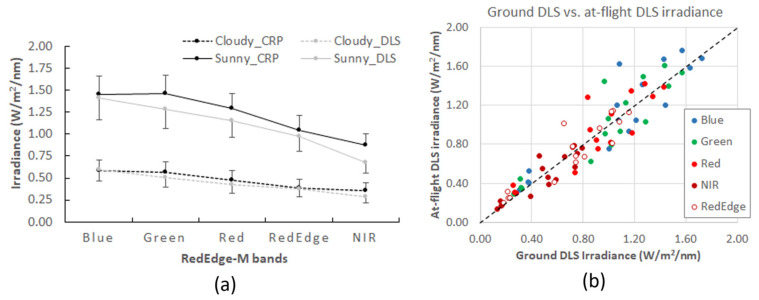
Sampled CRP and DLS irradiance curves on sunny and cloudy days (**a**) and comparison of on-ground vs. at-flight DLS irradiance during flight missions (**b**). The one standard deviation bar is marked above the CRP point and below the DLS point in (**a**).

**Figure 4 sensors-21-08224-f004:**
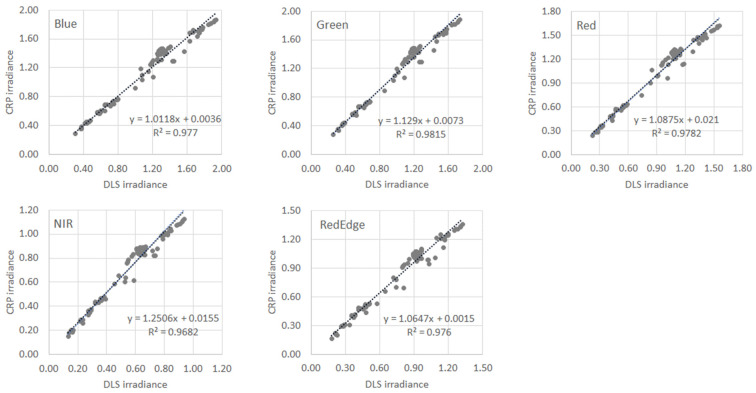
Scatterplots of the CRP and DLS irradiance of the five RedEdgeM bands. All calibration images in the study are utilized (N = 107).

**Figure 5 sensors-21-08224-f005:**
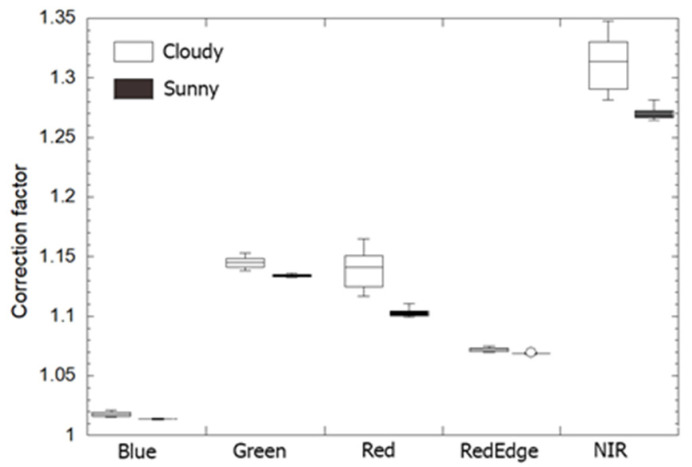
A boxplot of correction factors of the five RedEdge-M bands. All CRP calibration images in both field and drone experiments are used.

**Figure 6 sensors-21-08224-f006:**
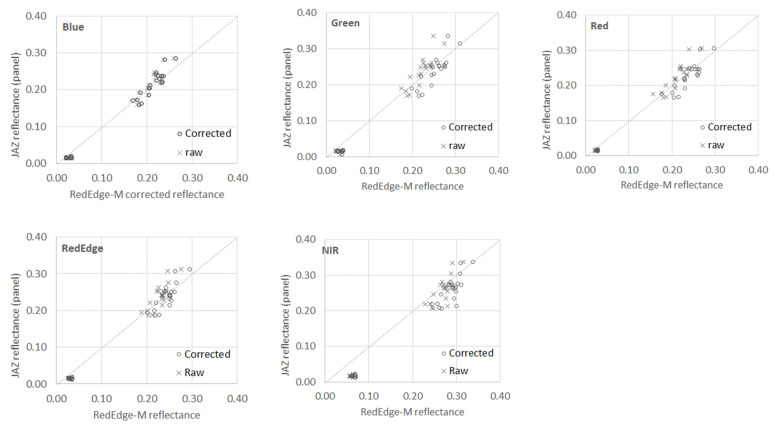
Comparison between the Jaz measured and the RedEdge-M reflectance of the three MAPIR panels: Black, dark grey, light grey. Both the raw and corrected reflectance values of the RedEdge-M camera are compared. All weather conditions are considered.

**Figure 7 sensors-21-08224-f007:**
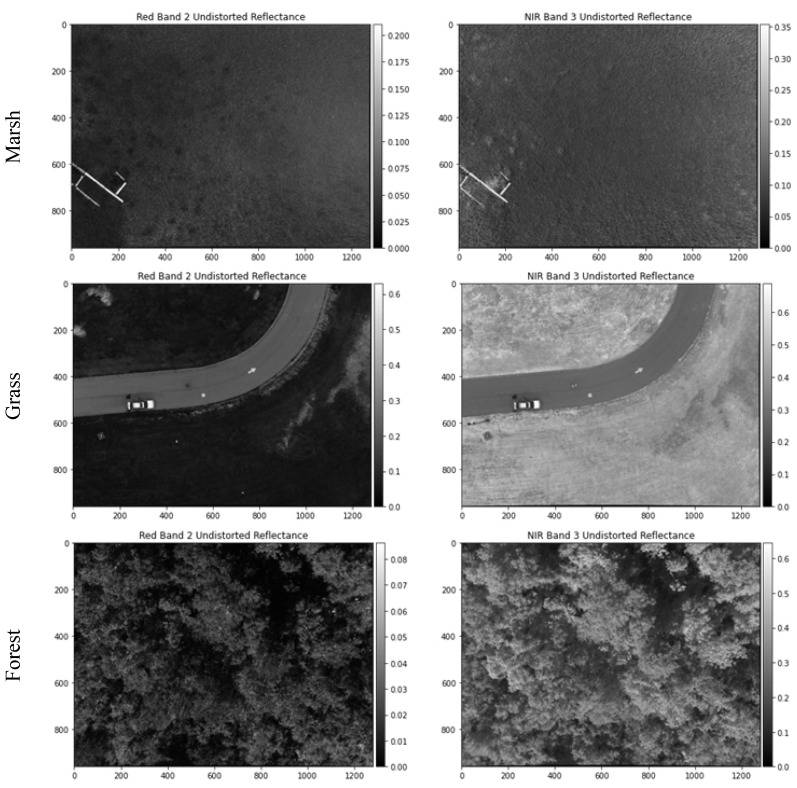
Example RedEdge-M surface reflectance images (marsh, grass, and forest) after at-sensor radiometric correction. Only red and NIR bands are displayed.

**Figure 8 sensors-21-08224-f008:**
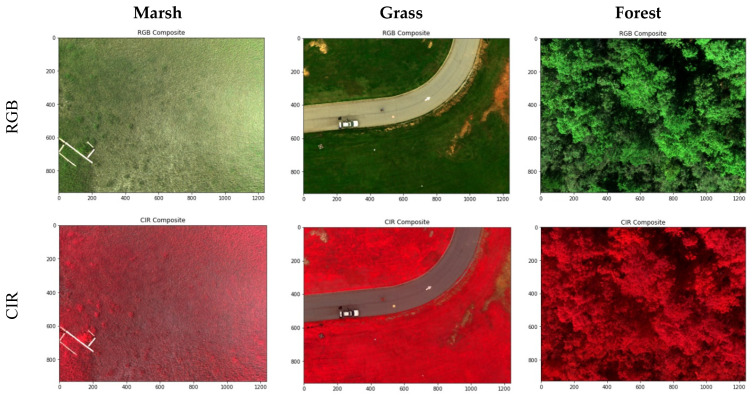
The normalization enhanced RGB and CIR composites of marsh, grass, and forest.

**Figure 9 sensors-21-08224-f009:**
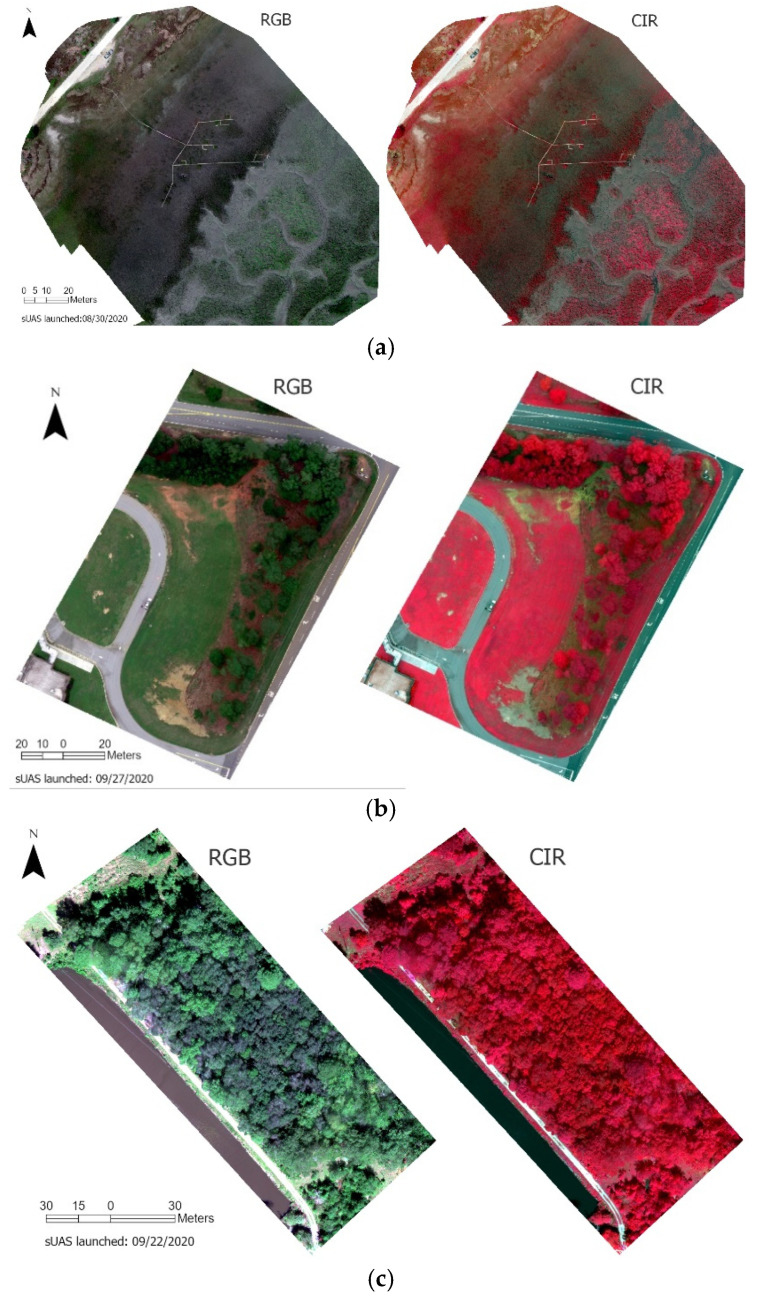
The RGB and CIR composites of the RedEdge-M surface reflectance orthoimages of marsh (**a**), grass (**b**), and forest (**c**).

**Figure 10 sensors-21-08224-f010:**
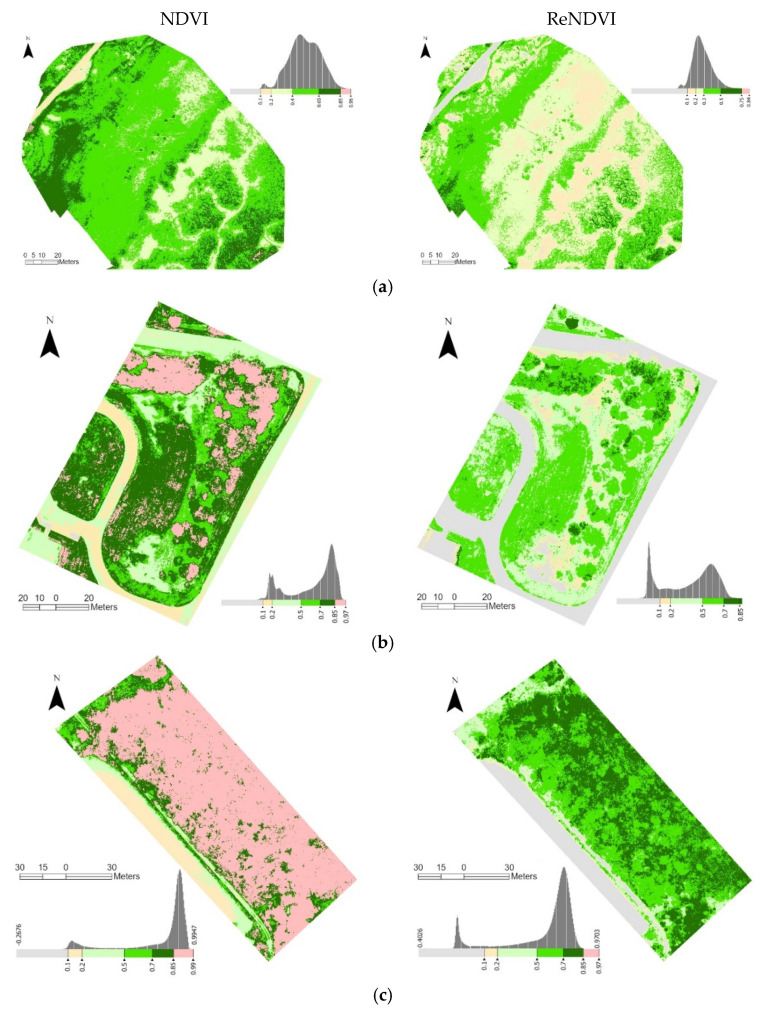
The NDVI and ReNDVI distributions of the RedEdge-M orthoimages for marsh (**a**), grass (**b**), and forest (**c**). The non-vegetation is effectively delineated with ReNDVI > 0.1.

**Table 1 sensors-21-08224-t001:** Spectral characteristics of the RedEdge-M camera and calibration panels.

Band Name	Central Wavelength (nm)	Bandwidth (nm)	Spectral Range (nm)	CRP Reflectance *	MAPIR Reflectance *			
					Black	Dark Gray	Light Gray	White
Blue	475	20	465–485	0.4893	0.0198	0.1880	0.2561	0.8269
Green	560	20	550–570	0.4895	0.0196	0.1974	0.2666	0.8722
Red	668	10	663–673	0.4899	0.0192	0.1935	0.2652	0.8772
Red Edge	717	10	712–722	0.4901	0.0194	0.2151	0.2670	0.8762
NIR	840	40	820–860	0.4905	0.0202	0.2334	0.2797	0.8668

* Note: The reflectance values are from the manufacturer (MicaSense and MAPIR).

**Table 2 sensors-21-08224-t002:** The list of experiments and collected samples (calibration images) to extract CRP and DLS irradiance. The three sites for flight missions are marsh, forest, and grass, respectively.

Experiments	Date	Weather	Time and Sample Size **				
On-ground	23 May 2021	Sunny	11:00 a.m.	11:15 a.m.	11:24 a.m.	11:42 a.m.	11:55 a.m.
			14	5	7	3	6
On-ground	30 May 2021	Cloudy	12:32 p.m.	12:40 p.m.	12:46 p.m.	12:52 p.m.	13:15 p.m. *
			7	10	8	14	15
Flight (marsh)	30 August 2020	Cloudy	Start Time		Sample size **		
			12:41 p.m.		2		
Flight (marsh)		Sunny/thin cloud	12:55 p.m.		2		
Flight (marsh)		Sunny/thin cloud	2:38 p.m.		1		
Flight (marsh)		Sunny/thin cloud	3:36 p.m.		2		
Flight (forest)	26 August 2020	Sunny/thin cloud	12:20 p.m.		2		
Flight (forest)	22 September 2020	Sunny/thin cloud	12:32 p.m.		2		
Flight (grass)	27 September 20	Cloudy	1:56 p.m.		2		
Flight (grass)	18 August 2020	Cloudy	2:18 p.m.		2		
Flight (grass)	4 August 2020	Sunny	10:23 a.m		2		

* It transited to a sunny (with cloud patches) condition. ** Sample size means the number of calibration images.

**Table 3 sensors-21-08224-t003:** Coefficients used to extract the correction factors of the five RedEdge-M bands.

	Blue	Green	Red	Red Edge	NIR
a	1.0118	1.1290	1.0875	1.0674	1.2506
b	0.0036	0.0073	0.0210	0.0015	0.0155
$\rho_{CRP}$	0.4893	0.4895	0.4899	0.4901	0.4905
$L_{CRP}$	mission-specific				

**Table 4 sensors-21-08224-t004:** Comparison of the pre-calibrated and field-measured reflectance of the MAPIR white and RedEdge-M CRP panels. For both panels, the pre-calibrated reflectance values are highlighted in bold.

	MAPIR White Panel							RedEdge-M CRP Panel			
	Pre-Calibrated (Jaz)	RedEdge-M Measurement (Raw)			RedEdge-M Measurement (Corrected)			Pre-Calibrated (RedEdge-M)	Jaz Measurement		
		All-Weather	Sunny	Overcast	All-Weather	Sunny	Overcast		All-Weather	Sunny	Overcast
Blue	**0.827**	0.694/0.041 *	0.723	0.652	0.705/0.041 *	0.733	0.664	**0.4893**	0.494/0.067 *	0.477	0.518
Green	**0.872**	0.728/0.047	0.760	0.679	0.828/0.051	0.863	0.776	**0.4895**	0.505/0.063	0.493	0.522
Red	**0.877**	0.722/0.051	0.758	0.667	0.807/0.047	0.838	0.761	**0.4899**	0.478/0.071	0.459	0.506
Red Edge	**0.876**	0.713/0.055	0.753	0.655	0.763/0.058	0.805	0.701	**0.4901**	0.477/0.070	0.458	0.506
NIR	**0.867**	0.707/0.051	0.744	0.652	0.757/0.054	0.796	0.399	**0.4905**	0.483/0.063	0.466	0.510

* Both the average and standard deviation values are marked in this column.

**Table 5 sensors-21-08224-t005:** Correction factors of the five RedEdge-M bands in three missions (marsh, grass, forest).

Site	Date	Weather	Blue	Green	Red	Red Edge	NIR
Marsh	30 August 2020	Sunny/thin cloud	1.0141	1.1343	1.1041	1.0690	1.2732
Grass	27 September 2020	Cloudy	1.0223	1.1541	1.1754	1.0755	1.3657
Forest	22 September 2020	Sunny	1.0146	1.1354	1.1079	1.0690	1.2746

**Table 6 sensors-21-08224-t006:** The min-max ranges of the corrected reflectance images for marsh, grass, and forest. The extreme values are highlighted in bold.

	Reflectance	Blue	Green	Red	Red Edge	NIR
Marsh	Min	0.0098	0.0192	0.0159	0.0321	0.0616
	Max	0.1397	0.2104	0.2116	0.2551	0.4110
Grass	Min	0.0235	0.0647	0.0321	0.0899	0.1473
	Max	0.4423	0.6098	0.6440	0.6312	0.8356
Forest	Min	**0.0009**	**0.0018**	**0.0004**	0.0074	0.0587
	Max	0.0525	0.1200	0.0666	**0.2636**	**0.6762**

## Data Availability

Not applicable.

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
