# Peer review of "At-Sensor Radiometric Correction of a Multispectral Camera (RedEdge) for sUAS Vegetation Mapping"

_sensors, 2021, doi:10.3390/s21248224_

Round 1
Reviewer 1 Report
In the manuscript, the author tests the performance of an RedEdge-M camera in extracting surface reflectance orthoimages and vegetation index products for drone-assisted vegetation mapping. An at-sensor radiometric correction procedure is established by interactively utilizing the CRP calibration panel and the built-in DLS sensor. This study is important for the radiometric correction of cameras on UAS. I suggest the study should pay more attention on the method of radiometric correction of cameras for UAS and to give more practical experimental suggestions, but not to analyze the images of each fields. Below are my specific comments:
Line 7: Please spell out the sUAS at the first time in the text.
Line 84: I suggest giving the map of the study site locations.
Line 136, a punctuation is required.
Line 163, the expression of the sentence is not accurate that the reflective property won’t change, but the factors can impact the measured results.
Line 198, please check Eq. 8. Is it right? I deduced a different result based on the other equations you gave.
Line 212, I suggest that section 3.4 can move to section 2, because this part introduces the experiment. Even the part from line 267 to 292 should be put in section 2, which are related with the experiment.
Line 229, in Table 2, what is “sample size”?
Line 236, do you mean Eq.8 instead of Eq.7?
Line 240: Why sUAS flights were not launched on these two days? Please give the reason.
Line 274, why is there an “O’ shape in the MAPRI panel? Is there any influence on the radiometric correction?
Line 384-387, Jaz field spectra are taken as true values, why does it varied at different measurements? What is the true values?
Line 388 Figure 6: The corrected RedEdge-M reflectance by both your at-sensor radiometric correction algorithm and utilizing the CRP panel itself should be compared with the Jaz measured reflectance.
Line 411, section 4.4 pays more attention on analyzing and comparing the images, however more quantitative results are required, such as the comparison of the results applying the old and new radiation correction method on the images. Some typical spectra should be compared quantitatively instead of the color of images.
Line 531, section 5 can be merged with section 6. It is short and there are no important contents of the discussion about the radiometric correction.
Author Response
Please see the attached response letter.

Reviewer 2 Report
In this article, a methodology for radiometric correction of a MicaSense RedEdge camera is presented.
Through several works, a radiometric correction model for the sensor has been created. These works have been carried out on sites with different vegetation, in order to make an accurate calibration.
In my opinion, I think that the article is well structured and the correct steps have been followed to make the conclusions. However, I don't know to what extent this work is certainly original...
There are many articles that deal with the same thing, for example:
https://www.mdpi.com/1424-8220/19/20/4453/htm
The author hardly cites any works on the subject. It would be advisable for him to do an in-depth search on the subject and improve the introduction considerably.
Once he improves the introduction and is aware of all that is out there, he can possibly mount a better argued discussion, as he does not take into account works by other authors on the subject.
I propose a major revision. The author needs to greatly improve the introduction with works by other authors and greatly improve the discussion.
Author Response
Responses to Reviewer #2:
In this article, a methodology for radiometric correction of a MicaSense RedEdge camera is presented. Through several works, a radiometric correction model for the sensor has been created. These works have been carried out on sites with different vegetation, in order to make an accurate calibration. In my opinion, I think that the article is well structured and the correct steps have been followed to make the conclusions. However, I don't know to what extent this work is certainly original...
Response: Thanks for raising the concern. I fully agree that the sections of “Introduction” and “Discussion” were not comprehensively outlined. The revised manuscript made major revision on these two sections. Details are as below. Additionally, a new sub-section “2.2 Flight and on-ground experiments” (Line 140-172) is added to give more practical description of the experimental designs. All revisions are highlighted in red in the revised manuscript.
- There are many articles that deal with the same thing, for example: https://www.mdpi.com/1424-8220/19/20/4453/htm. The author hardly cites any works on the subject. It would be advisable for him to do an in-depth search on the subject and improve the introduction considerably.
Thanks for recommending this reference. This article deals with the RedEdge cameras that my study is testing. From this one, I made further literature review. A total of nine articles are added to support my study.
In the “Introduction”, new descriptions are added to introduce past studies about new drone sensors such as a hyperspectral camera (Line 54-56). The need of calibration of these sensors (Line 57-61) and past studies about in-lab calibration (Line 61-65). The one you suggested is cited here as an in-lab calibration example.
A new paragraph is followed to introduce the in-situ calibration conducted real time during sUAS flight (Line 67-76). As reported in past studies, high percent errors remained. This is connected to the research goal of my study.
Hope the revised structure of the “Introduction” fairly explains the research need and originality of this study.
- Once he improves the introduction and is aware of all that is out there, he can possibly mount a better argued discussion, as he does not take into account works by other authors on the subject.
Exactly. Major revision was made in the section of “Discussion”. Several paragraphs were added in order to broaden and deepen the discussion. Line 557-569 discusses the advantage of this study from other RedEdge studies. Line 583-599 brought up the limitation of this study and the need of future work. Line 600-613 discussed the findings of NDVI overestimation by RedEdge cameras as support by past studies. Finally, I made further discussion on the proposed ReNDVI including its difference from the current NDRE and the agreement with other studies (Line 614-626). The last paragraph indicates the applicability of the corrected RedEge camera in future satellite remote sensing (Line 635-639).
Hope the revised “Discussion” section provides better justification of this study’s outcomes and contribution to current literature.
Reviewer 3 Report
The study described in this manuscript is well-explained and written. These site-specific corrections allow users to apply a 'customized' common approach for analyzing the landscape which, at the very least, provides a good starting point for an analysis.
While reading the manuscript, I had hoped the author would demonstrate the impact of the proposed at-sensor radiometric correction model by comparing the vegetation indices based on the traditional correction methods (as implemented in Pix4D or the MicaSense GitHub libraries) and the author's proposed adjustment. Without that comparison, I am left wondering if the unexpectedly high NDVI values (Figure 11) are due to the author's at-sensor adjustments. I think a comparison along these lines would provide additional value to those who might consider using the author's approach of atmospheric correction in their work.
This manuscript does, however, does provide the community (those of us who use vegetation indices in our work) a useful framework for generating consistent data collected with our high-resolution, multispectral sensors.
I have few specific remarks regarding the document itself...
267: I suggest modifying “When taking the RedEdge-M images of the CRP panel, solar radiation could be..” to “When acquiring the RedEdge-M images of the CRP panel, solar radiation can be..”
286: I suggest changing “…Even in sunny” to “…Even on sunny”
297: I suggest changing “…, then cloud gradually reduced.” To “…, then clouds gradually reduced”
311-313: Please add caption for panel (b) in Figure 3
380: “…MAPIR) were compared…”
501: It is unclear what the author means by “… the NDVI turns to be overestimated,…”
Author Response
Responses to Reviewer #3:
The study described in this manuscript is well-explained and written. These site-specific corrections allow users to apply a 'customized' common approach for analyzing the landscape which, at the very least, provides a good starting point for an analysis. While reading the manuscript, I had hoped the author would demonstrate the impact of the proposed at-sensor radiometric correction model by comparing the vegetation indices based on the traditional correction methods (as implemented in Pix4D or the MicaSense GitHub libraries) and the author's proposed adjustment. Without that comparison, I am left wondering if the unexpectedly high NDVI values (Figure 11) are due to the author's at-sensor adjustments. I think a comparison along these lines would provide additional value to those who might consider using the author's approach of atmospheric correction in their work.
Response: Very meaningful and important comment! I have taken it for granted per my long-time experience in bio-environmental remote sensing. To justify my statement about the overestimation of the RedEdge cameras, the results are compared with past studies using the same camera types. A new paragraph is added in the Discussion (Line 602-613):
“…This finding agrees with past studies utilizing the same cameras. For example, in an sUAS experiment for RedEdge calibration [20,15], the RedEdge imagery was collected at a flight height of 375ft on November 2, 2017 in Henrietta, NY, USA. The reflectance-extracted NDVI in a roadside grass field, similar as our grass site in this study, was in a range of 0.8 – 0.9, and was higher in cloudy than sunny days [20]. Be aware that the imagery was acquired at the end of growing season in a mid-latitude area (43.04°N, 77.70°W). The NDVI values apparently overestimate the greenness of natural vegetation. The overestimation performance was similar for RedEdge-3 and RedEdge-M [15] cameras. It is not a big concern when the end users visually compare the NDVI differences of a field in one single flight, as demonstrated in MicaSense brochure [11]. However, it is not practical to directly utilize the RedEdge NDVI for quantitative vegetation mapping.”
This manuscript does, however, does provide the community (those of us who use vegetation indices in our work) a useful framework for generating consistent data collected with our high-resolution, multispectral sensors.
Response: Thanks for the positive comment and encouragement. The following outlines the correction per your specific remarks.
- 267: I suggest modifying “When taking the RedEdge-M images of the CRP panel, solar radiation could be..” to “When acquiringthe RedEdge-M images of the CRP panel, solar radiation can.”
Corrected. Thanks.
- 286: I suggest changing “…Even in sunny” to “…Even onsunny”
Corrected.
- 297: I suggest changing “…, then cloud gradually reduced.” To “…, then cloudsgradually reduced”
Corrected.
- 311-313: Please add caption for panel (b) in Figure 3.
Corrected.
- 380: “…MAPIR) werecompared…”
Corrected. I also added the word “values” to make it more reasonable sentence: “… reflectance values … were compared…”
- 501: It is unclear what the author means by “… the NDVI turns to be overestimated,…”
The rest of paragraph explains in detail about the overestimation problem. It was revised as (Line 507-508):
“…At all three sites, the NDVI values are much higher than what we expect in natural environment…”
I highly appreciate your suggestions on grammatical editing. I also read through and corrected the errors across the manuscript (e.g., “on sunny days” instead of “in sunny days”).
Round 2
Reviewer 1 Report
According to the response and the new manuscript, there are still some issues in the manuscript.
- I deduced the equation (7) and got a different style (see the attachment). So the equation (8) should be different with yours.
These two equations are important for this study. Please check them carefully.
- Line416-418, why is there the highest variation vertically (CRP) or horizontally (MAPIR white) in Fig.6? An explanation is required.
- About the question (13) mentioned last time, I don’t think your explanation is reasonable. At-sensor radiometric correction for sUAS is the only one and the most important purpose of this manuscript. If the new method can’t be quantitatively proved its advantage and is not better than the traditional method, why do you develop it and why do the others want to use it?
In summary, I think the author should be more careful to revise the manuscript and it is necessary to increase more worthy information in this study.

Author Response
The equations cannot be pasted here. Please see the responses in the attached file.

Reviewer 2 Report
The author has improved the publication considerably. I think that with all these changes, the article has improved enormously.
At least the introduction now shows enough background and with that he has been able to set up a better discussion.
I agree with the changes that have been proposed by the other reviewers and to which the author has also answered correctly.
Therefore, I consider the article to be accepted.
Author Response
The author is grateful to the reviewer's support.